# Soil Bacterial Diversity and Potential Functions Are Regulated by Long-Term Conservation Tillage and Straw Mulching

**DOI:** 10.3390/microorganisms8060836

**Published:** 2020-06-02

**Authors:** Chang Liu, Lingling Li, Junhong Xie, Jeffrey A. Coulter, Renzhi Zhang, Zhuzhu Luo, Liqun Cai, Linlin Wang, Subramaniam Gopalakrishnan

**Affiliations:** 1Gansu Provincial Key Laboratory of Aridland Crop Science, Gansu Agricultural University, Lanzhou 730070, China; liuc@gsau.edu.cn (C.L.); xiejh@gsau.edu.cn (J.X.); zhangrz@gsau.edu.cn (R.Z.); luozz@gsau.edu.cn (Z.L.); cailq@gsau.edu.cn (L.C.); wangll@gsau.edu.cn (L.W.); 2College of Agronomy, Gansu Agricultural University, Lanzhou 730070, China; 3Department of Agronomy and Plant Genetics, University of Minnesota, St. Paul, MN 55108, USA; jeffcoulter@umn.edu; 4College of Resource and Environment, Gansu Agricultural University, Lanzhou 730070, China; 5International Crops Research Institute for the Semi-Arid Tropics (ICRISAT), Patancheru, Hyderabad, Telangana 502324, India; s.gopalakrishnan@cgiar.org

**Keywords:** conservation tillage, field pea, soil microbial community, high-throughput sequencing, PICRUSt

## Abstract

Soil physiochemical properties are regulated by cropping practices, but little is known about how tillage influences soil microbial community diversity and functions. Here, we assessed soil bacterial community assembly and functional profiles in relation to tillage. Soils, collected in 2018 from a 17-year field experiment in northwestern China, were analyzed using high-throughput sequencing and the PICRUSt approach. The taxonomic diversity of bacterial communities was dominated primarily by the phyla *Proteobacteria* (32–56%), *Bacteroidetes* (12–33%), and *Actinobacteria* (17–27%). Alpha diversity (Chao1, Shannon, Simpson, and operational taxonomic unit (OTU) richness) was highest under no-tillage with crop residue removed (NT). Crop residue retention on the soil surface (NTS) or incorporated into soil (TS) promoted the abundance of *Proteobacteria* by 16 to 74% as compared to conventional tillage (T). Tillage practices mainly affected the pathways of soil metabolism, genetic information processing, and environmental information processing. Soil organic C and NH_4_–N were the principal contributors to the diversity and composition of soil microbiota, whereas soil pH, total nitrogen, total P, and moisture had little effect. Our results suggest that long-term conservation practices with no-tillage and crop residue retention shape soil bacterial community composition through modifying soil physicochemical properties and promoting the metabolic function of soil microbiomes.

## 1. Introduction

Soil health has been increasingly recognized as one of the primary indicators of the sustainability of natural and managed ecosystems [1,2,3]. A key area of improving soil health is to understand the relationship among soil biological parameters, soil physiochemical properties, and land management practices [3,4]. In agroecosystems, the health of soil is closely linked to the structure, diversity, and function of soil microbiomes, as they are highly sensitive to environmental conditions and cropping practices [5,6,7]. Furthermore, soil microorganisms play an important role in nutrient cycling and fertility maintenance of the soil, which is a key factor influencing the resilience and productivity of agroecosystems [8,9,10].

Conservation practices, such as no-tillage and crop residue retention on the soil surface, offer significant benefits in conserving soil and water, increasing soil organic matter, and boosting crop productivity [11,12]. These practices can alter the physical and chemical properties of soils, leading to changes in the composition and metabolic capabilities of soil microbial communities [13,14]. However, conventional farmers in the Loess Plateau of China traditionally remove crop residues out of the field at harvest for animal feed or household heating sources, and moldboard plowing is typically performed for land preparation; this leads to soil fertility losses and land degradation [15]. Some farmers have recently adopted more diversified crop rotation systems with N-fixing crops such as dry pea (*Pisum sativum* L.) rotated with cereals such as wheat (*Triticum aestivum* L.). There is a need to determine how different tillage and straw retention practices in the cereal–legume rotation systems influence soil microbial communities.

A number of short-term (typically 2 to 5 years) studies have shown that conservation practices play a crucial role in influencing the diversity of soil microorganisms [16,17]. However, short-term field experiments may only reveal the temporary consequences of different cropping practices on soil microbial communities, and long-term field experiments are needed to help understand the biological mechanisms responsible for the impact of cropping practices on the potential function of soil microorganisms. Some recent studies have addressed the long-term impact of tillage practice on soil microbial diversity and metabolic functions [18,19,20]. In the case of the semi-arid Loess Plateau of western China where the loss of soil fertility and land degradation has been a serious issue [21], some published studies have mostly determined the taxonomic composition and phylogenetic diversity of microbial communities. Although such information is highly valuable to help understand the diversity of soil microbial community, a fundamental need is to investigate the potential association between the metabolic function of soil microbiomes and farming practices.

The bioinformatics approach, PICRUSt (phylogenetic investigation of communities by reconstruction of unobserved states), has recently become available for the determination of the metabolic and functional profiles in a broad range of host-associated microbial communities [22]. This computational technique can predict the functional composition of a metagenome based on 16S rRNA gene profile and reference genome database using an extended ancestral-state reconstruction algorithm. This ‘predictive metagenomic’ approach, albeit imperfect in many cases, can provide insights into uncultivated microbial communities [22]. Some studies have used PICRUSt analysis to compare microbiomes’ presumptive functions between different soil profiles [23,24,25]. However, it is unknown whether this approach can be used to determine the variation of microbial community functional capabilities under different tillage and crop residue management practices.

In the present study, we determined the association of microbial community functional capabilities with different tillage and crop residue management practices in a long-term field experiment that has been run at the Loess Plateau of western China for the past 18 years (since 2001). Bacterial community diversity and predictive functions were determined using 16S rRNA amplicon sequencing and PICRUSt analysis (PICRUSt v.1). We hypothesized that the composition of soil microbial community and the potential functions are regulated by tillage and crop residue management practices, and these effects are related to the change of soil physiochemical properties. The specific objectives were to (1) determine the effect of tillage and crop residue management practices on soil physiochemical traits, (2) characterize the taxonomic distribution, phylogenic composition, and predictive functional profiles of the soil bacterial communities, and (3) evaluate the variation and changes in the abundance, composition, and potential functions of soil microbiomes under different tillage and crop residue management practices.

## 2. Materials and Methods

### 2.1. Site Description

The study was conducted at the Rainfed Agricultural Experimental Station (35°28′ N, 104°44′ E; 1971 m a.s.l.) of Gansu Agricultural University, Gansu Province, northwest China. The station is at the temperate semi-arid zone in the western Loess Plateau, with a long-term (2001–2019) average solar radiation of 5.67 KJ m^−2^ and an annual sunshine duration of 2477 h. The mean annual air temperature is 6.4 °C, with accumulated temperature above 10 °C of 2339 °C, and a frost-free period of 140 d. Mean annual precipitation is 391 mm year^−1^, occurring mainly between June and September. The soil at the experimental site is classified as a Calcaric Cambisol (FAO/UNESCO 1990). In the 0–30 cm soil depth, it has a pH of 8.36, and a sandy loam texture with up to 50% sand. Before 2001, the field site had a long history of conventional tillage practice with flax (*Linum usitatissimum* L.) as the main crop.

### 2.2. Experimental Design

The long-term experiment, initiated in 2001, had four tillage and crop residue management treatments that were applied to both phases of a two-year spring wheat–field pea rotation in a randomized completely block design with three replicates. The four treatments were no-tillage with all crop residue removed at harvest (NT), no-tillage with crop residue chopped and spread evenly on the soil surface (NTS), conventional tillage with all crop residue removed at harvest (T), and conventional tillage with all crop residue chopped, spread evenly on the soil surface, and incorporated into the soil via plowing (TS). Tillage practices in the TS and T treatments included moldboard plowing to a depth of 20 cm in the fall, followed by harrowing prior to planting in the spring. Each year, spring wheat (cv. Dingxi No. 40) was sown in late March and harvested in late July to early August, and field pea (cv. Lvnong No. 2) was planted in early April and harvested in early July. Plots were 4 m wide × 20 m long. The seeding rate was 187.5 and 100 kg ha^−1^ for spring wheat and field pea, respectively, with a row spacing of 20 and 24 cm, respectively. Nitrogen (N) fertilizer in the form of urea (46% N) was applied at a rate a 105 and 20 kg N ha^−1^ for spring wheat and field pea, respectively. Phosphorus fertilizer in the form of calcium superphosphate (6% P_2_O_5_) was applied at a rate of 46 kg P_2_O_5_ ha^−1^. All N and phosphorus fertilizers were applied at sowing.

### 2.3. Soil Sampling and Physicochemical Properties Analysis

Soil samples from field pea plots were collected at flowering on 17 June 2018, 86 days after pea was sown. Five cores of soil were randomly obtained from each plot from the 0–20 cm depth using a 5-cm diameter soil corer to provide a composite sample. Samples were bulked, homogenized, and sieved through a 2-mm mesh to remove rocks and surface litter. Each composite sample was split into two subsamples; one was stored immediately at −80 °C for DNA extraction, and the other was stored at −4 °C for soil physicochemical properties analysis.

Soil pH was measured in a deionized soil suspension with soil:water ratio of 1:2.5 (mass: volume) using a pH meter (Mettler Toledo FE20, Shanghai, China). Soil organic carbon (SOC) and total nitrogen (TN) were determined using a modified Walkley–Black wet oxidation method and Kjeldahl method, respectively. Olsen phosphorus was extracted with 0.5 M NaHCO_3_ and measured using the colorimetric method. Both soil NH_4_^+^–N and NO_3_^−^–N were extracted with 2.0 M KCl and measured using a UV-1800 spectrophotometer (Mapada Instruments, Shanghai, China). The soil moisture was assessed gravimetrically.

### 2.4. DNA Extraction, PCR Amplification, and Illumina Sequencing

Total DNA was extracted from approximately 1.5 g of fresh soil thrice (0.5 g sample for each time) using an E.Z.N.A.^®^ soil DNA kit (Omega Bio-Tek, Inc., Norcross, GA, USA) according to the manufacturer’s protocol. The extracted DNA concentration was detected using a NanoDrop-2000 spectrophotometer (Thermo Fisher Scientific, Wilmington, DE, USA). The PCRs were performed with the following procedures: 2 min of initial denaturation at 98 °C, followed by 25 cycles of 15 s at 98 °C, 30 s for annealing at 55 °C, 30 s for elongation at 72 °C, and a final extension at 72 °C for 5 min. The PCR components contained 5 μL of Q5 reaction buffer (5×), 5 μL of Q5 High-Fidelity GC buffer (5×), 0.25 μL of Q5 High-Fidelity DNA Polymerase (5 U/μL), 2 μL (2.5 mM) of dNTPs, 1 μL (10 uM) of each Forward and Reverse primer, 2 μL of DNA Template, and 8.75 μL of ddH_2_O. PCR amplifications were conducted with the universal primers 343F (5′-TACGGRAGGCAGCAG-3′) and 798R (5′-AGGGTATCTAATCCT-3′) targeting the V3–V4 hypervariable region of the bacterial 16S rRNA gene [26]. High-throughput sequencing was performed on the Illumina Miseq platforms at OE BioPharm Technology Co., Ltd., Shanghai, China. All raw sequencing data (.fg files) were deposited in the NCBI Sequence Read Archive (SRA) database under accession number PRJNA627529.

### 2.5. Data Processing and Bioinformatics Analysis

To reduce the negative effects of random sequencing errors, we used Trimmomatic software to detect and remove low quality sequences in paired-end reads [27]. After trimming, paired-end reads were assembled using FLASH software [28]. Sequences were clustered to generate operational taxonomic units (OTUs) using a UPARSE pipeline with 97% similarity [29]. The Quantitative Insights Into Microbial Ecology (QIIME) software package (version 1.8.0) was used to analyze the sequence reads [30]. All representative reads were annotated and blasted against the Silva database (Version 123) using the RDP classifier with a confidence threshold of 70% [31].

### 2.6. Statistical Analysis

Taxonomic-based alpha diversity analysis was carried out using the QIIME software package (version 1.8.0) to calculate community richness and evenness based on the Chao1, Shannon Wiener, and Simpson diversity indexes. Beta diversity analysis was performed based on the UniFrac distance matrices, which were generated by analysis of similarities and permutation multivariate analysis of variance (PERMANOVA) with 999 permutations. Principal coordinate analysis (PCoA) based on the weighted and unweighted UniFrac distance was employed to illustrate the clustering of the different samples for taxonomic and phylogenetic community comparison using the “vegan” package (version 2.5-5) in R (version 3.6.1) statistical software (R Foundation for Statistical Computing, Vienna, Austria). The prediction of metagenomic functions of soil microbiomes was performed using the PICRUSt pipeline [22]. Functional counts and annotations were assigned and compared with the Kyoto encyclopedia of genes and genomes (KEGG) Orthology values (level 1, 2, and 3) to produce functional counts × samples tables [32]. To assess the impact of soil physiochemical properties on the distribution of dominant phyla and the clustering of soil samples, redundancy analysis was conducted using CANOCO software (version 5.0, Microcomputer Power, Inc., Ithaca, NY, USA). Analysis of variance was performed using SPSS software (version 21.0; SPSS Inc., Chicago, IL, USA) to assess the effect of the treatments on microbial diversity indexes and significance between treatments was detected using Tukey’s HSD test at *p* < 0.05, unless otherwise noted.

## 3. Results

### 3.1. Soil physicochemical Properties

Soil physiochemical properties were analyzed in 2018, which was 17 years after the treatments had been in place (since 2001). Soil organic carbon, TN, TP, and NO_3_–N differed significantly among the treatments (*p* < 0.05) (Table 1). Soil organic carbon in the NTS treatments was significantly greater (*p* = 0.038) than that in the T treatment and ranked as NTS > NT > TS. There was no significant difference in soil moisture and pH value among the treatments (*p* > 0.05). Soil TN ranged from 0.83 to 0.96 g kg^−1^ with highest concentration of TN obtained in the NTS treatment. Soil NO_3_–N was lowest with the TS treatment, and TP with the NT treatment was significantly greater (*p* = 0.033) than with the TS treatment.

### 3.2. Sequencing Depth and Alpha Diversity

Using Illumina MiSeq sequencing of 16S rRNA gene amplicons, a total of 485,373 filtered sequences remained after quality control and 400,824 reads (ranging from 31,342 to 36,654 reads per sample) were generated for further bioinformatic analysis (Appendix A). All these sequences were subsequently clustered into 5738 OTUs based on 97% similarity. The number of observed OTUs detected in each individual sample ranged from 1909 to 2926. The Good’s coverage index of each sample was >0.975, and the rarefaction curves were close to the saturation phase, indicating that sufficient sequencing coverage was achieved and that the OTUs were representative of the overall microbial community libraries.

There were significant differences among the treatments in the alpha diversity of bacterial populations, as shown by Chao1, Simpson, Shannon, and OTU richness alpha diversity indexes (Figure 1). Among treatments, NT soil samples had highest Chao1, Shannon, Simpson, and OTU richness indexes, suggesting that the NT treatment resulted in greater richness of bacterial populations than the other treatments. The Shannon and Simpson indexes for the NT (*p* = 0.007) and TS (*p* = 0.020) treatment were significantly higher than that for the T treatment, but no significant difference was observed between the NTS and T treatments (*p* > 0.05).

### 3.3. Beta Diversity and Microbial Community Composition

The PCoA results showed dissimilarity in bacterial community structure across the different treatments. The total variance in bacterial community structure explained by PCo1 and PCo2 was 65.33% and 19.07%, respectively, based on the weighted UniFrac distance (Figure 2a), and 33.45% and 12.27%, respectively, based on the unweighted UniFrac distance (Figure 2b). There was one major cluster comprising samples from the NT and TS treatments, whereas the samples from the NTS and T treatments had distinct patterns in community composition, which were separated from the major cluster.

The taxonomic diversity of bacterial communities across soil samples was dominated primarily by the phyla *Proteobacteria* (32.21–56.15%), *Bacteroidetes* (12.38–32.96%), and *Actinobacteria* (17.09–27.38%), which accounted for 77.51–89.09% of the relative abundance of bacterial communities, followed by the phyla *Firmicutes*, *Gemmatimonadetes,* and *Acidobacteria*, with low relative abundance (Figure 3, Appendix A). Soil bacterial community structure in the NT and TS treatments had a similar trend. In comparison with the T treatment, the NTS, NT, and TS treatments increased the abundance level of the phylum *Proteobacteria* by 74.33%, 15.84%, and 19.53%, respectively. The *Proteobacterial* phylum could be assigned to four classes, *α-proteobacteria*, *β-proteobacteria*, *γ-proteobacteria*, and *δ-proteobacteria*. In comparison with the T treatment, the three conservation practices NTS, NT, and TS increased the abundance level of the class *α-proteobacteria* by 18.6%, 63.5%, and 52.6%, respectively; and similarly, NT and TS treatments increased the abundance of *γ-proteobacteria* by 14.1% and 23.3%. Furthermore, there were three assigned classes: *Bacteroidia*, *Flavobacteria*, and *Sphingobacteria*, in the phylum *Bacteroidetes*. Compared to the T treatment, the NTS treatment increased the microbial community abundance by 47.2% at *Flavobacteria* and 34.8% at the *Sphingobacteria* class level (Appendix A).

### 3.4. Variation of Predicted Functions of Soil Microbial Community

The PICRUSt approach was applied to gain insight into the detailed metabolic and functional profiles of soil bacterial community assembly at level one (Table 2) and level two (Table 3). At level one (Table 2), the weighted nearest sequenced taxon index (NSTI) score for the evaluation of prediction accuracy across all samples ranged from 0.078 to 0.131, with a mean of 0.112 ± 0.015. The PICRUSt analysis identified seven level one KEGG Orthology groups (KOs). Among the seven functional categories at level one, the metabolism, genetic information processing, and environmental information processing were the dominant functional groups, accounting for 51.4%, 15.8%, and 13.9% of total abundance of functional genes, respectively. The predicted functional profiles corresponding to level one KEGG pathways across soil samples are depicted in Figure 4, where the soils under NT had the highest level of metabolism, organismal systems, environmental information processing, and genetic information processing, among the treatments.

At level two of the functional profiles of soil bacterial community, a total of 41 KEGG pathways were found, and nine groups showed significant differences among the tillage and crop residue management treatments (Table 3). At level two, the NT treatment had the significantly (*p* < 0.01) high enrichment in the KEGG subcategories of lipid metabolism, amino acid metabolism, and xenobiotics biodegradation and metabolism, while the T treatment had highest enrichment in the subcategories of cellular processes and signaling, glycan biosynthesis and metabolism, nucleotide metabolism, and enzyme families. The NT treatment increased lipid metabolism by 8.6% and amino acid metabolism by 5.6% compared to the T treatment. The TS treatment decreased glycan biosynthesis and metabolism by 17.8% compared to the T treatment. The functional groups including genetic information processing and metabolic diseases were significantly enriched with the NTS and T treatments (*p* < 0.05). Seven of the subcategories belonging to genetic information processing and environmental information processing showed no significant difference among the treatments (*p* > 0.05).

### 3.5. Relationships between Soil Parameters and Microbial Community Structure

The relationships between key soil physicochemical properties and the structure of bacterial communities at the phylum level were illustrated by the direction and length of the vectors (Figure 5). The redundancy analysis showed that axes 1 and 2 explained 75.30% and 5.56% of the total variance in soil bacterial community composition, respectively. The phyla *Actinobacteria*, *Chloroflexi*, *Gemmatimonadetes*, *Acidobacteria*, and *Nitrospirae* were clustered together to the edge of soil NO_3_–N, and negatively correlated with soil TN and moisture. Among the soil physiochemical properties, SOC explained 52.0% of the variance (*F* = 10.8, *p* = 0.002) and NH_4_-N explained 11.2% of the variance (*F* = 3.8, *p* = 0.048), both being the most significant predictors across soil samples (with 999 permutations) (Appendix A). The abundance and diversity of the phylum *Bacteroidetes* was correlated with NH_4_–N, while *Proteobacteria* was highly correlated with SOC and bacterial community structures in the NTS treatment.

## 4. Discussion

Soil bacterial community assemblies, one of the key soil health indicators [4,33], are often shaped by biotic and abiotic factors including plant type [34], farming practice [33], and environmental conditions [16,35,36]. In the present study, we characterized the structure and functional profiling of soil bacterial communities and compared the effects of different tillage and crop residue management practices on the microbiomes in the eighteenth year of a cereal-legume rotation. Short-term (3–5 years, etc.) field experiments can help determine the temporary change of soil microbial community in relation to farming practices [34,37,38], while long-term (>10 years) field experiments may allow researchers to learn what would happen in the soil environment under different farming practices. Our results from the 18 years of soil treatments (tillage, crop residue management) demonstrated that cropping practices had a significant effect on soil physiochemical properties and the composition of soil microbiomes. These effects explained much of the variation in the metabolic and functional profiles of soil bacterial communities. In particular, no-tillage with crop residue retention on the soil surface had a strong, positive effect on the richness and evenness of soil bacterial communities; this suggests that the adoption of conservation soil management practices has great potential to enhance the health of the soil via promoting soil microbial biodiversity and richness.

In the present study, the effect of soil management practices on the bacterial community structure and functionality was closely associated with soil physiochemical properties. Among them, SOC and NH_4_–N were the highest responsible indicators of the functionality of bacterial communities under different soil management practices. Although SOC, TN, TP, and NO_3_–N varied significantly among the treatments, the highest differences in SOC and TN were observed between the T and NTS practices. Previous studies have indicated that long-term no-tillage can improve the physical, chemical, and biological properties of the soil in different agroecosystems [39,40,41], while crop residue retention can enhance SOC sequestration through reducing potential negative effects caused by fluctuation in temperature and moisture in the upper soil profile [42,43]. However, many studies report controversial results on the effect of tillage on SOC, largely because of short- and medium-term tillage treatments [44,45]. It is well-known that crop residue decomposition in the soil require a long period of time before the organic matter is detectible [44,46]. In addition, variable carbon concentration in the material of different crop types may contribute to the controversial effect on SOC in response to soil management practices. More diversified cropping systems with cereal–oilseed–legume rotations improve soil N availability [8] and enhance soil microbial biodiversity [8,47]. In our study, the combination of tillage and crop residue retention had a significant effect on soil physiochemical properties, which is most likely because these soil management practices influenced the composition of particulate soil organic matter and nutrient cycling. This partly explains the results that the amounts of SOC, TN, and TP in the 0–20 cm soil layer were higher under treatments with crop residue retention than treatments with residue removed.

Alpha diversity of soil bacterial community was assessed using Chao1, Shannon, Simpson, and OTU richness indexes. The greatest bacterial diversity was found in the NT treatment, while all alpha diversity indexes for the NTS treatment were higher than those with the T treatment, suggesting that no-tillage combined with crop residue retention favors the recruitment of specific groups of soil organisms and increases substrate availability, therefore enhancing bacterial diversity [48]. The analysis for the relative abundance of microbial taxonomic components showed that Proteobacteria was the most predominant bacterial phylum in our study, followed by Bacteroidetes and Actinobacteria; similar results have been reported by others [49,50,51]. Our study quantified that the NTS, NT, and TS treatments increased the abundance of the phylum Proteobacteria by 74%, 16%, and 20%, respectively, compared with conventional tillage practice.

To elucidate soil environmental factors that are potentially associated with shift in the abundance of major soil bacterial phyla, researchers proposed a copiotroph–oligotroph concept in an ecological context for soil bacteria [52]. The phylum *Proteobacteria* are described as fast-growing copiotrophs, which preferentially consume labile SOC pools and maintain great nutritional availability when soil environmental conditions favor the growth of soil microbiota [53]. No-tillage is shown to modify the soil profile with improved ventilation, leading to greater nutrient availability than conventional tillage; this was probably the main reason that higher relative abundances of *Proteobacteria* and Bacteroidetes were observed under no-tillage plus crop residue retention treatments than in conventional tillage in our study. Bacteria belonging to the *Acidobacteria* phylum are considered oligotrophs [54], which exhibit slow growth rates in soils where organic C quality and/or quantity is high [53]; this explains why the relative abundance of *Acidobacteria* was lower than copiotrophic groups in our study. In addition, the *α-proteobacteria* class was detected with higher relative abundance across the samples in the present study. Given that this class is described as heterotrophic, nitrogen-fixing microbes, higher TN in the conservation tillage treatments might be the reason for the higher proportion of this bacterial class [55]. Reduced tillage practices can enhance SOC and improve soil aggregate stability [39], thereby increasing the relative abundance of profitable functional bacteria groups.

Many studies have focused on the biochemical and structural properties, metabolic pathways, gene regulation, and evolutionary history of soil microbiomes to provide insights into the linkage between the structure of microbial communities and their function [12,56,57]. The occurrences of functional genes observed in our experiment could be explained by the relative abundance of certain microbial categories in the cycling of major biological elements (i.e., C, N, and P) and other biological activities. This is supported by the observations that the KEGG pathways of the metabolism of amino acid, terpenoids, and polyketides, and the biosynthesis of other secondary metabolites were significantly higher in the NT treatment compared with the T treatment. *Proteobacteria* and *Actinobacteria* represented a great proportion of the bacterial community, which are commonly detected in biological carbon and nitrogen cycling process. Additionally, N-fixing microbes such as *Azotobacter* and *Burkholderia*, belonging to *Proteobacteria*, carry out nitrogen fixation, dissimilatory nitrate reduction to ammonia, nitrification, and denitrification in soil ecosystems [58]. Likewise, a wide variety of heterotrophic bacteria, especially those belonging to *Actinobacteria* and *Proteobacteria*, degrade soluble organic molecules such as organic acids, amino acids, sugars, and even recalcitrant carbon compounds [59,60].

With the rapid growth in the number of sequenced genomes, the PICRUSt tool has been increasingly used to infer functions that are likely associated with a marker gene based on its sequence similarity with a reference genome [61,62,63,64]. In agroecosystem studies, this approach has been used to explore the relationship between soil microbial communities and farming practices by characterizing metabolic and functional capabilities of the communities across a broad range of host-associated and environmental samples [65,66,67]. In the present study, we employed the PICRUSt approach and inferred the two major functional categories corresponding to the metabolism and genetic information processing on KEGG pathways. We found significant impacts on the tillage and crop residue management practices on KEGG pathways associated with soil bacterial communities. These findings strengthen our understanding of how different soil management practices impacted the functional properties of soil bacterial communities. However, the PICRUSt approach has its intrinsic limitations [22]. Marker-gene sequencing does not provide direct information about the functional genes that are present in the genomes of community members. The phylogenetic information contained in 16S rRNA marker gene sequences may not be sufficiently well correlated with genomic content to generate a high level of predictions. In our analysis, only 16S rRNA marker gene sequences corresponding to bacterial genomes were included, and the system was unable to infer viral or eukaryotic components of a metagenome. In addition, the ‘potential function’ of the bacterial community predicted by PICRUSt may have included those spore form, inactive bacteria present in the soil. Therefore, the accuracy of the ‘functional property’ of soil bacterial communities predicted by this approach requires further investigation. Furthermore, the ‘predictive function’ may be questionable in the sense that the approach ignores the pangenome concept. There is a need to extensively evaluate the merits and limitations of PICRUSt in the prediction of functional capacities of soil microbiomes. It is desirable that improved methods can be developed in the near future, so that more accurate predictions of functional potential of soil microbial communities can be generated.

## 5. Conclusions

In the present study, we found that 17 years of continuous tillage and crop residue management practices in a cereal–legume rotation significantly altered soil physiochemical properties such as SOC, TN, and TP, and therefore modified the composition and functional profile of soil bacterial communities. No-tillage treatments significantly increased the richness and evenness of soil bacterial populations compared to conventional tillage practices. The functional shift in the soil bacterial community under different tillage systems was largely related to changes in metabolism and genetic information processing, while SOC and NH_4_–N served as the significant predictors for soil bacterial community shift. These findings provide deep insights into the role of farming practices in shaping soil bacterial communities and their functions in agroecosystems.

## Figures and Tables

**Figure 1 microorganisms-08-00836-f001:**
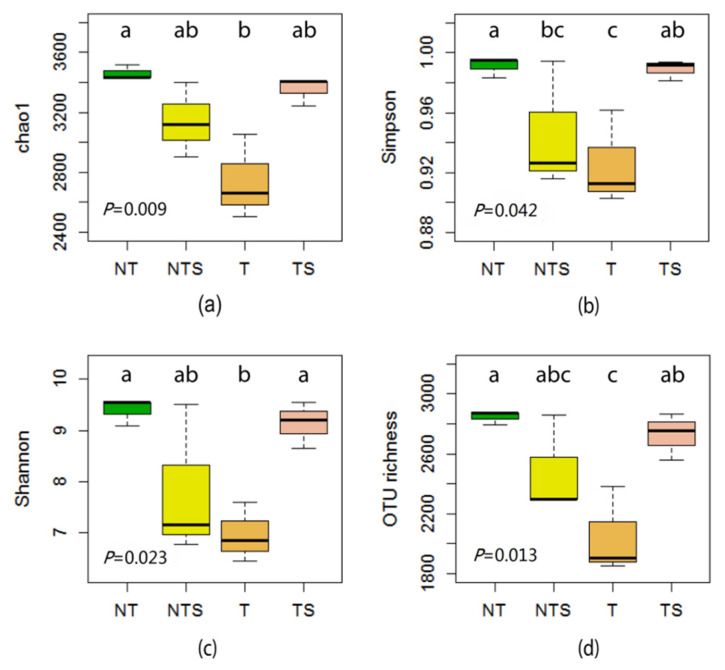
Alpha diversity of the soil bacterial community according to the Chao1 (**a**), Simpson (**b**), Shannon (**c**), and OTU richness (**d**) indexes as affected by tillage and crop residue management treatments. NT, no-tillage with all crop residue removed at harvest; NTS, no-tillage with crop residue chopped and spread evenly on the soil surface; T, conventional tillage with all crop residue removed at harvest; TS, conventional tillage with all crop residue chopped, spread evenly on the soil surface, and incorporated into the soil via plowing. Boxplots with different letters above the boxes denote means that are significantly different (*p* < 0.05).

**Figure 2 microorganisms-08-00836-f002:**
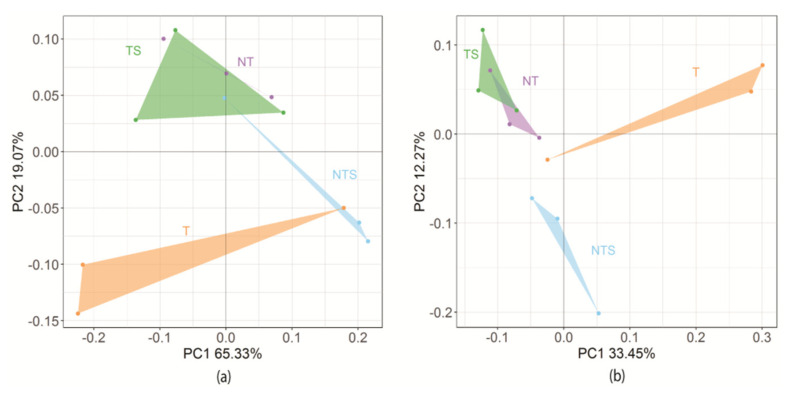
Summary of principal coordinate analysis of soil bacterial composition as affected by tillage and crop residue management treatments based on (**a**) the weighted UniFrac distance and (**b**) unweighted UniFrac distance. NT, no-tillage with all crop residue removed at harvest; NTS, no-tillage with crop residue chopped and spread evenly on the soil surface; T, conventional tillage with all crop residue removed at harvest; TS, conventional tillage with all crop residue chopped, spread evenly on the soil surface, and incorporated into the soil via plowing.

**Figure 3 microorganisms-08-00836-f003:**
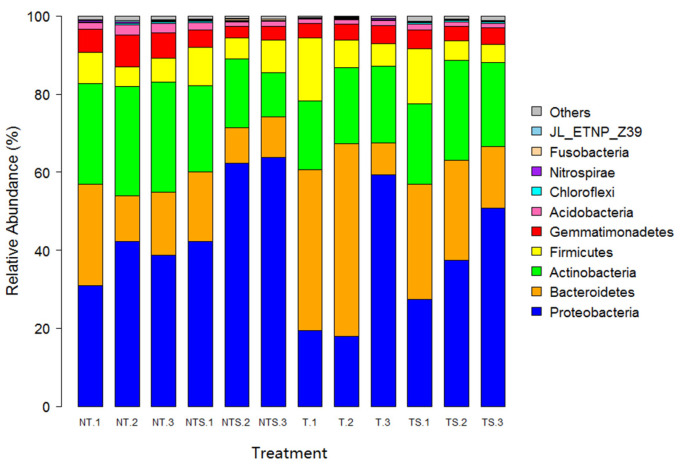
Relative abundance of top 10 soil bacterial phyla for all samples as affected by tillage and crop residue management treatments. NT, no-tillage with all crop residue removed at harvest; NTS, no-tillage with crop residue chopped and spread evenly on the soil surface; T, conventional tillage with all crop residue removed at harvest; TS, conventional tillage with all crop residue chopped, spread evenly on the soil surface, and incorporated into the soil via plowing. The numbers following the treatment name denote the sampling replications. For example, NT1, NT2, and NT3 means the soil sampling was taken from replicate 1, 2, and 3 of the field plots, respectively.

**Figure 4 microorganisms-08-00836-f004:**
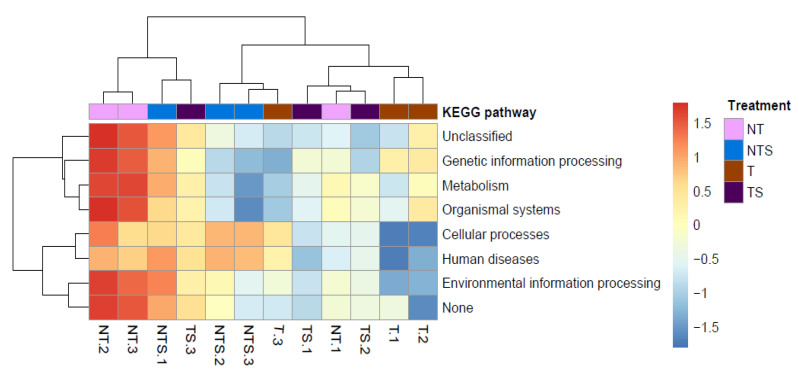
Clusters showing the relative abundance of soil bacterial community predicted functions related to KEGG pathways at level one as affected by tillage and crop residue management treatments. NT, no-tillage with all crop residue removed at harvest; NTS, no-tillage with crop residue chopped and spread evenly on the soil surface; T, conventional tillage with all crop residue removed at harvest; TS, conventional tillage with all crop residue chopped, spread evenly on the soil surface, and incorporated into the soil via plowing.

**Figure 5 microorganisms-08-00836-f005:**
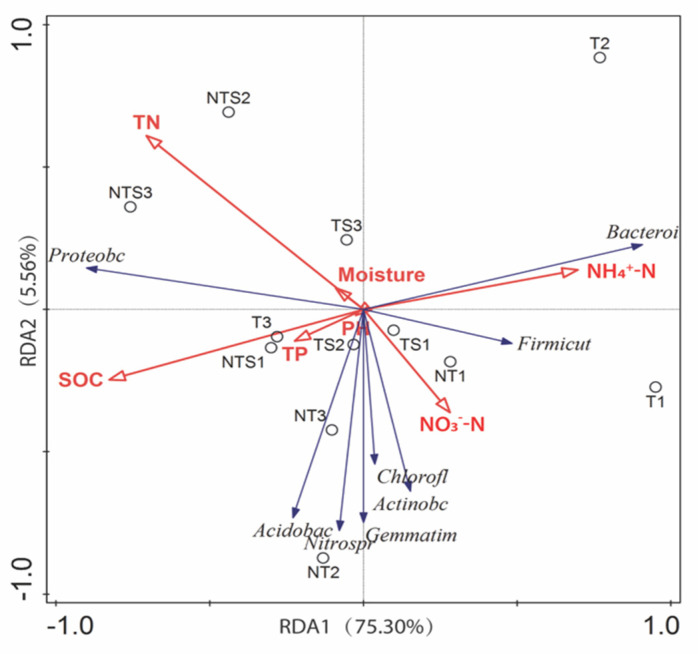
Summary of redundancy analysis, showing the relationships between soil parameters and soil bacterial community structure. Red lines represent soil parameters, blue lines represent the bacterial phylum-level taxonomy, and empty circles represent soil samples from all replications (*n* = 3) of each tillage and crop residue management treatment. SOC, soil organic carbon; TN, total nitrogen; TP, total phosphorus; NT, no-tillage with all crop residue removed at harvest; NTS, no-tillage with crop residue chopped and spread evenly on the soil surface; T, conventional tillage with all crop residue removed at harvest; TS, conventional tillage with all crop residue chopped, spread evenly on the soil surface, and incorporated into the soil via plowing. Numbers following treatment abbreviations denote the experimental replication.

**Table 1 microorganisms-08-00836-t001:** Soil physiochemical properties as affected by tillage and crop residue management treatments.

Soil Parameter ^a^	Treatment ^b^			
	NT	NTS	T	TS
pH	8.12 ± 0.10 a ^c^	8.07 ± 0.08 a	8.07 ± 0.03 a	8.02 ± 0.12 a
Moisture (%)	12.21 ± 1.02 a	12.16 ± 0.33 a	12.54 ± 1.13 a	12.03 ± 0.37 a
SOC (g kg^−1^)	13.07 ± 0.25 ab	13.20 ± 0.26 a	12.50 ± 0.53 b	13.03 ± 0.25 ab
TN (g kg^−1^)	0.83 ± 0.05 b	0.96 ± 0.06 a	0.84 ± 0.08 b	0.85 ± 0.05 b
TP (g kg^−1^)	0.77 ± 0.15 a	0.73 ± 0.07 ab	0.71 ± 0.14 ab	0.52 ± 0.13 b
NO_3_–N (mg kg^−1^)	38.53 ± 2.21 a	32.26 ± 2.29 ab	36.24 ± 2.38 ab	27.73 ± 9.48 b
NH_4_–N (mg kg^−1^)	1.41 ± 0.44 a	1.13 ± 0.34 a	1.49 ± 0.50 a	1.37 ± 0.23 a

^a^ SOC, soil organic carbon; TN, total nitrogen; TP, total phosphorus. ^b^ NT, no-tillage with all crop residue removed at harvest; NTS, no-tillage with crop residue chopped and spread evenly on the soil surface; T, conventional tillage with all crop residue removed at harvest; TS, conventional tillage with all crop residue chopped, spread evenly on the soil surface, and incorporated into the soil via plowing. ^c^ Within a row, data (means ± SD, *n* = 3) followed by different letters are significantly different (*p* < 0.05).

**Table 2 microorganisms-08-00836-t002:** Relative abundance of soil bacterial community predicted functions related to Kyoto encyclopedia of genes and genomes (KEGG) pathways at level one [32] as affected by tillage and crop residue management treatments.

KEGG Pathway	Treatment ^a^			
	NT	NTS	T	TS
Environmental information processing	13.74 ± 0.06 a ^b^	14.49 ± 0.36 a	13.44 ± 1.12 a	13.78 ± 0.17 a
Genetic information processing	15.58 ± 0.06 a	15.59 ± 0.23 a	16.41 ± 0.99 a	15.68 ± 0.60 a
Metabolism	52.28 ± 0.36 a	50.09 ± 1.45 b	51.40 ± 1.20 ab	51.97 ± 0.66 ab
Organismal systems	0.862 ± 0.003 a	0.817 ± 0.018 b	0.863 ± 0.036 a	0.852 ± 0.010 ab
Cellular processes	3.51 ± 0.21 a	4.16 ± 0.47 a	3.24 ± 0.82 a	3.51 ± 0.20 a
Human diseases	0.92 ± 0.02 b	1.11 ± 0.09 a	0.91 ± 0.15 b	0.94 ± 0.07 ab
Unclassified	12.90 ± 0.10 b	13.55 ± 0.044 a	13.54 ± 0.16 a	13.08 ± 0.33 ab
None	0.198 ± 0.002 a	0.205 ± 0.005 a	0.193 ± 0.019 a	0.199 ± 0.006 a

^a^ NT, no-tillage with all crop residue removed at harvest; NTS, no-tillage with crop residue chopped and spread evenly on the soil surface; T, conventional tillage with all crop residue removed at harvest; TS, conventional tillage with all crop residue chopped, spread evenly on the soil surface, and incorporated into the soil via plowing. ^b^ Within a row, data (means ± SD, *n* = 3) followed by different letters are significantly different (*p* < 0.05).

**Table 3 microorganisms-08-00836-t003:** Relative abundance of soil bacterial community top four predicted functions related to KEGG pathways at level two as affected by tillage and crop residue management treatments.

KEGG Pathway	Treatment ^a^				
	*p*-Value	NT	NTS	T	TS
**Metabolism**					
Carbohydrate metabolism	0.256	10.62 ± 0.08 a ^b^	10.29 ± 0.21 a	10.79 ± 0.48 a	10.56 ± 0.17 a
Energy metabolism	0.342	5.60 ± 0.05 a	5.46 ± 0.18 a	5.67 ± 0.19 a	5.62 ± 0.07 a
Lipid metabolism	0.030	4.17 ± 0.01 a	3.89 ± 0.15 bc	3.84 ± 0.12 c	4.09 ± 0.14 ab
Nucleotide metabolism	0.136	3.21 ± 0.02 a	3.21 ± 0.04 a	3.45 ± 0.23 a	3.24 ± 0.12 a
Amino acid metabolism	0.027	11.18 ± 0.12 a	10.59 ± 0.36 b	10.59 ± 0.13 b	11.09 ± 0.25 a
Metabolism of other amino acids	0.071	2.04 ± 0.03 a	1.92 ± 0.09 a	1.91 ± 0.02 a	2.03 ± 0.08 a
Glycan biosynthesis and metabolism	0.079	1.77 ± 0.03 a	1.86 ± 0.09 a	2.19 ± 0.33 a	1.80 ± 0.15 a
Metabolism of cofactors and vitamins	0.398	4.20 ± 0.03 a	4.15 ± 0.05 a	4.26 ± 0.13 a	4.20 ± 0.02 a
Metabolism of terpenoids and polyketides	0.084	2.45 ± 0.04 a	2.26 ± 0.16 a	2.25 ± 0.02 a	2.41 ± 0.11 a
Biosynthesis of other secondary metabolites	0.118	1.03 ± 0.01 a	0.92 ± 0.07 a	1.00 ± 0.08 a	1.01 ± 0.02 a
Xenobiotics biodegradation and metabolism	0.044	4.19 ± 0.06 a	3.68 ± 0.35 ab	3.52 ± 0.22 b	4.09 ± 0.33 a
**Genetic information processing**					
Transcription	0.317	2.46 ± 0.01 a	2.50 ± 0.05 a	2.52 ± 0.06 a	2.46 ± 0.05 a
Translation	0.478	4.12 ± 0.03 a	4.09 ± 0.08 a	4.34 ± 0.35 a	4.16 ± 0.19 a
Folding, sorting and degradation	0.170	2.19 ± 0.01 a	2.25 ± 0.06 a	2.31 ± 0.09 a	2.20 ± 0.06 a
Replication and repair	0.286	6.85 ± 0.03 a	6.79 ± 0.09 a	7.27 ± 0.54 a	6.89 ± 0.30 a
**Environmental information processing**					
Membrane transport	0.380	11.56 ± 0.07 a	11.99 ± 0.13 a	11.33 ± 0.86 a	11.61 ± 0.16 a
Signal transduction	0.182	2.03 ± 0.09 a	2.32 ± 0.23 a	1.94 ± 0.31 a	2.01 ± 0.08 a
Signaling molecules and interaction	0.283	0.181 ± 0.004 a	0.193 ± 0.015 a	0.198 ± 0.017 a	0.182 ± 0.007 a
**Unclassified**					
Poorly characterized	0.446	5.00 ± 0.04 a	5.12 ± 0.13 a	5.06 ± 0.06 a	5.01 ± 0.10 a
Metabolic diseases	0.027	.081 ± 0.001 b	.089 ± 0.006 a	.091 ± 0.004 a	.082 ± 0.002 b
Metabolism	0.431	2.48 ± 0.03 a	2.49 ± 0.01 a	2.53 ± 0.07 a	2.53 ± 0.04 a
Genetic information processing	0.101	2.16 ± 0.01 a	2.29 ± 0.06 a	2.27 ± 0.09 a	2.20 ± 0.06 a
Enzyme families	0.117	1.93 ± 0.02 a	1.95 ± 0.04 a	2.03 ± 0.07 a	1.93 ± 0.06 a
Cellular processes and signaling	0.028	3.28 ± 0.07 c	3.68 ± 0.26 ab	3.70 ± 0.07 a	3.37 ± 0.17 bc

^a^ NT, no-tillage with all crop residue removed at harvest; NTS, no-tillage with crop residue chopped and spread evenly on the soil surface; T, conventional tillage with all crop residue removed at harvest; TS, conventional tillage with all crop residue chopped, spread evenly on the soil surface, and incorporated into the soil via plowing. ^b^ Within a row, data (means ± SD, *n* = 3) followed by different letters are significantly different (*p* < 0.05).

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
