# Peer review of "Soil Bacterial Diversity and Potential Functions Are Regulated by Long-Term Conservation Tillage and Straw Mulching"

_microorganisms, 2020, doi:10.3390/microorganisms8060836_

Round 1

Reviewer 1 Report

Dear Authors,

the paper is interesting. Microbiome sequencing is a recognised method that provides a lot of information. A widely accepted and recognized method of high-throughput sequencing was used. I have objections to the use of Picrust, but I have written about it below.

However, there are several shortcomings in the paper. Below are some comments and please read them.

Abstract

Too long. According to the instructions, it should have max 200 words, this has 255.

Introduction

L63-75 – I don't know if it's a lack of knowledge of literature, or if it was only about research in China. Long-term research is conducted all over the world. Including conventional crops:

Gajda et al. 2019 Soil Research 57(2) 124-131 https://doi.org/10.1071/SR18113; Valboa et al. 2015 Soil and Tillage Research 154 126-135 https://doi.org/10.1016/j.still.2015.06.017; Govaerts et al. 2007 Applied Soil Ecology 37 18-30 https://doi.org/10.1016/j.apsoil.2007.03.006.

As regards the absence of such research in the area covered by the manuscript, please rewrite this paragraph to make it clear. However, referring to only one particular region will make the work purely local in nature, and this will reduce its impact.

Materials and Methods

Where is the description of soil physicochemical properties analysis? They are given in the results, but no methodology is given.

L143-145 – What was PCR conditions? Add literature for primers.

L149/158 – It seems to me that the “data processing” and “bioinformatics analysis” should be together. Or separate the “statistical analysis” and “bioinformatics analysis”, on two paragraph.

L167-169 - Why wasn't PIRCUSt2 used?

Results

Fig 1, 2, 3, 5 – Small font, illegible. Please increase the size of the subtitles a little.

L248 – “data not shown” can be included as a supplement.

Fig 3 – “top 10” in each sample or for all samples? please specify,

L290, 305 – What does +/-? How much was “n?” Add the test that was used.

Unify P<0.05 - once is italic and once is not.

Discussion

L394-404 - there is no critical approach to the method used. PICRUSt it is a tool with great limitations when it comes to soil samples; the developers write: “16S rRNA gene sequences do not typically enable resolution of strain variation within a species. Strains of prokaryotic species can vary in gene content to remarkable degrees and horizontal gene transfer can frequently occur between distantly related taxa, so the predictions should always be taken with a grain of salt.” and “A related issue is that the certain environments are better represented by reference genomes than others. For instance, PICRUSt2 is expected to perform better on 16S sequences from the human gut compared to the cow rumen, even if the actual 16S sequences themselves are very similar.” – how it comes for soil environment? https://github.com/picrust/picrust2/wiki/Key-Limitations.

In my opinion, there is a lack of references to analysis that used this tool for soil research (is only one – ref. 52). There are also bacteria in the soil in spore form, inactive - the sequencing will detect them in the sample, but the functions they can potentially perform are inactive at a given moment, so we cannot talk about the influence of treatment on the function, because we are not sure that the given functions are active at that moment. 

PICRUSt may be used to analyze non-human microbiomes, but this is only a "prediction" of the presence of the genes responsible for these functions and not an unambiguous identification of the functions.

That's all my comments.
Please answer/correction.

Author Response

Reviewer #1:

Abstract

  1. Response to comment: The abstract of this manuscript is too long. According to the instructions, it should have max 200 words, this has 255.

Response:  The abstract has been shortened to 200 words in the revision without changing the main scientific points found from this study (L19-34).

Introduction

  1. Response to comment: L63-75 –Long-term research is conducted all over the world and there are some of reference papers including conventional crops are focused on long term tillage impact on soil microbial diversity.

Response: Following your instruction and suggestions, we have added some relevant references in the revised manuscript and modified those statements accordingly (L59-71).

Materials and Methods

  1. Response to comment: Soil physicochemical properties analysis is given in the results, but no methodology is given.

Response: L155, 162-168-We have added the description of soil physicochemical properties into “Materials and Methods” section according to the reviewer’s suggestions.

  1. Response to comment: Provide the information about PCR conditions and add literature for primers.

Response: L174-179- We have provided detailed information about PCR conditions and components in this experiment; we also added the literature for the primer as the reviewer suggested.

  1. Response to comment: L149/158- “data processing” and “bioinformatics analysis” should be put together; Or separate the “statistical analysis” and “bioinformatics analysis”, on two paragraphs.

Response: L185, 194- We have modified the titles of two paragraphs and separated the “bioinformatics analysis” and “statistical analysis” as reviewer suggested.

  1. Response to comment: Why wasn't PIRCUSt2 used?

Response: This is an excellent point. PICRUSt2 analysis methodology has become available a while ago and it should have been used, however, we had a ‘time issue”; it was unfortunate we had done the analysis before the PICRUSt2 was commonly available. As you might agree with us that this type of the analysis takes a lot of effort, time, and resource. We believe that the results from the PICRUSt1, albeit not being superior as PICRUSt2, are still valuable in term of meeting the objectives of the project we are studying.

Results

  1. Response to comment: Fig 1, 2, 3, 5-Please increase the size of the subtitles a little.

Response: Fig 1, 2, 3, 5-We have modified the font of subtitles as reviewer suggested.

  1. Response to comment: L248 –“data not shown” can be included as a supplement.

Response: L297-We added figure S1 at the end of the certain sentence to replace the phrase “data not shown”.

  1. Response to comment: Fig 3 – “top 10” in each sample or for all samples? Please specify,

Response: L303- We have made specification with “for all samples” according to the Reviewer’s comments.

  1. Response to comment: L290, 305 – What does +/-? How much was “n?” Add the test that was used.

Response: L233,344, 359- We modified the description of means and SD with “n=3” if it was required.

  1. Response to comment: Unify P<0.05 - once is italic and once is not.

Response: L261- We are very sorry for our incorrect writing of “P<0.05”. We have made correction according to the Reviewer’s comments.

Discussion

  1. Response to comment: L394-404 - there is no critical approach to the method used. PICRUSt it is a tool with great limitations when it comes to soil samples; the developers write: “16S rRNA gene sequences do not typically enable resolution of strain variation within a species. Strains of prokaryotic species can vary in gene content to remarkable degrees and horizontal gene transfer can frequently occur between distantly related taxa, so the predictions should always be taken with a grain of salt.” and “A related issue is that the certain environments are better represented by reference genomes than others….Please check out the revision report for detailed information.

Response: We totally agree with the reviewer in regard to the advancement and the limitation of the methodology used in the study. We realized the limitations of the methodology as the original developers cautioned. In the scientific literature, there are a number of studies that have used the same/similar methodology for soil microbiome study and the resulted findings have been accepted by the world scientific communities. In order to satisfy the reviewer’s concerns on this, we have modified several statements in the Discussion (Lines 396-409) by adding a number of relevant references on the subject. The deficiency of the methodology in predicting the potential function of soil bacterial communities are briefly discussed. This addition has strengthened our points. We trust that the scientific findings from our study, albeit not superior and let alone perfect, still provide significant insights about how tillage practices influence soil microbial community diversity and metabolic functions. One of the unique features of our study is the soil samples from the long-term field experiment, and we believe that the publication of those novel findings from the long-term (18 years) treatments will have some significant, positive impacts on the understanding of the relationship between tillage and soil microbiomes.

Reviewer 2 Report

Overall this is a good manuscript with some interesting results from a such a long-term experiment. There are a few things I would like clarification on and/or changed. I would like to see the sample size within the statistical analysis stated explicitly, as I could not tell if the three DNA replicates were combined during the analysis and if so, how/when combination was performed. Three replicates per treatment is low, and it would have been nice if the multiple samples from each plot had been sequenced separately rather than bulked, however the authors have done well in analysing and interpreting the data within the sample size constraints. There is no description of the methods used for the soil physicochemical analyses which need to be added. The overall ANOVA p-values are not presented and should be included for interpretation purposes. Fisher’s protected LSD which was used for post-hoc comparison of the ANOVAs does not correct for multiple comparisons and a more conservative test such as Tukey’s HSD would be more appropriate. Which PICRUSt version was used and can this be stated in the text? If it was not v2 then it may need to be re-run as v1 assumes greengenes was used to assign taxonomy.

Line by line comments:

Line 25-26 (+30): The level of detail within the numbers (to two decimal places) is excessive, would recommend rounding to nearest whole number

Lines 31-33: This sentence needs revising – I think it should say that tillage has a larger impact on metabolism etc than other metabolic/functional groups?

Lines 57-58: “More educated farmers” is an unnecessary value judgement on what causes farmers to shift farming practices, I suggest changing to “Some farmers” unless you have a reference showing education is highly correlated with farming choices.

Lines 65-71: Grammar needs revising.

Line 137: No methods given for soil physicochemical property analysis.

Line 174: I assume should be “Analysis of variance”

Line 181-182: This interpretation seems incorrect, the letters in the table indicate that soil organic carbon is significantly greater in the NTS treatment compared to T but that the NT and TS are not significantly different from T.

Line 208: I am unsure what you mean by “observed species diversity” – is this OTU richness? Species shouldn’t be used in this context. If this is OTU richness please replace throughout, if not can the term be defined within the method.

Lines 210-212: This sentence does not make sense, please revise.

Lines 213-219: Are there only three replicates per treatment within Figure 1? If so, a dotplot or some other type of plot would give more information than a boxplot. Please add the ANOVA p-values to the figure legend or some appropriate place in the text.

Line 244: replace “improved” with “increased”.

Line 249: There is one till sample that is more similar to the other treatments, also apparent in the ordination plot – can you guess as to why this might be? Underlying gradients in soil properties across the plots?

Line 299: Can the p-values from the ANOVA be added to this table as a column for ease of interpretation?

Line 313: pH shows no trend with the community composition, TP and moisture show practically no trend, and so I don’t think they should be included in interpreting the relations between phyla and soil properties.

Line 316: Is this 63.2% number the result of adding together the individual SOC and NH4-N R2 values? If so, this procedure will not work if there is any correlation between SOC and NH4-N so I would recommend removing the 63.2% number.

Line 344-6: Unclear what you mean by “somewhat distinct patterns” – they have similar diversity, I also don’t think you can conclude anything about the health of the soil based upon purely microbial richness.

Line 367: NTS had intermediate bacterial diversity, NT had the highest bacterial diversity. This sentence needs revising as based on incorrect conclusion.

Line 397: The PICRUSt authors suggest < 0.06 as a good match (and > 0.15 as a high one), I would suggest rephrasing that your matches were reasonable.

Author Response

Reviewer #2:

  1. Response to comment: Line 25-26 (+30): The level of detail within the numbers (to two decimal places) is excessive, would recommend rounding to nearest whole number

Response: L24, 27-We have modified the data in form of whole number in “Abstract” section according to the reviewer’s recommendation.

  1. Response to comment: Lines 31-33: This sentence needs revising – I think it should say that tillage has a larger impact on metabolism etc than other metabolic/functional groups?

Response: L28-29- We have revised the sentence and made it more clear and complete as the reviewer suggested.

  1. Response to comment: Lines 57-58: “More educated farmers” is an unnecessary value judgement on what causes farmers to shift farming practices, I suggest changing to “Some farmers” unless you have a reference showing education is highly correlated with farming choices.

Response: L53-We have made correction with the phrase “Some farmers” according to the reviewer’s comments.

  1. Response to comment: Lines 65-71: Grammar needs revising.

Response: Those sentences have been modified to make the point much clearer.

  1. Response to comment: Line 137: No methods given for soil physicochemical property analysis.

Response: L155, 162-168-We have added the detailed description of soil physicochemical properties analysis into “Materials and Methods” section according to the reviewer’s suggestions.

  1. Response to comment: Line 174: I assume should be “Analysis of variance”

Response: L214- We are very sorry for our incorrect writing and have made correction.

  1. Response to comment: Line 181-182: This interpretation seems incorrect; the letters in the table indicate that soil organic carbon is significantly greater in the NTS treatment compared to T but that the NT and TS are not significantly different from T.

Response: L221-222- we have revised the sentence and corrected the mistake which the reviewer pointed out.

  1. Response to comment: Line 208: I am unsure what you mean by “observed species diversity” – is this OTU richness? Species shouldn’t be used in this context. If this is OTU richness please replace throughout, if not can the term be defined within the method.

Response: L247, 250,252-the “observed species diversity” index represents the OTU amounts detected in each sample. The high observed species diversity index refers to high microbial species abundance. It is really true as reviewer suggested that “OTU richness” would be more accurate than “observed species diversity” in alpha diversity analysis. We have replaced the phrase “observed species diversity” instead of using “OTU richness” in the manuscript and figure.

  1. Response to comment: Lines 210-212: This sentence does not make sense, please revise.

Response: L252-253- we have revised the sentence and corrected the mistakes which the reviewer pointed out.

  1. Response to comment: Lines 213-219: Are there only three replicates per treatment within Figure 1? If so, a dotplot or some other type of plot would give more information than a boxplot. Please add the ANOVA p-values to the figure legend or some appropriate place in the text.

Response: L254-We have performed Tukey’s HSD analysis and added the ANOVA p=values to each figure which represents alpha diversity index in Fig. 1.

  1. Response to comment: Line 293: replace “improved” with “increased”.

Response: L293-We have replaced “improved” with “increased” as the reviewer suggested.

  1. Response to comment: Line 249: There is one till sample that is more similar to the other treatments, also apparent in the ordination plot – can you guess as to why this might be? Underlying gradients in soil properties across the plots?

Response: We have tried to figure out why it is, but it is unfortunate that we did not find out an appropriate answer to your concern. Sorry for this.

  1. Response to comment: Line 299: Can the p-values from the ANOVA be added to this table as a column for ease of interpretation?

Response: L353-354-We have performed Tukey’s HSD analysis for the relative abundance of top four predicted functions related to KEGG pathways at level two and added the ANOVA p=values to the table as a column in Table 3.

  1. Response to comment: Line 313: pH shows no trend with the community composition, TP and moisture show practically no trend, and so I don’t think they should be included in interpreting the relations between phyla and soil properties.

Response: L365-367- It is true as reviewer mentioned that some of soil physicochemical properties show no trend with the community composition, so we describe the result without pH, TP and moisture factors to explain the relationship between microbial composition and soil properties.

  1. Response to comment: Line 316: Is this 63.2% number the result of adding together the individual SOC and NH4-N R2 values? If so, this procedure will not work if there is any correlation between SOC and NH4-N so I would recommend removing the 63.2% number.

Response: L370-we have removed the 63.2% number and modified the sentence as the reviewer suggested.

  1. Response to comment: Line 344-6: Unclear what you mean by “somewhat distinct patterns” – they have similar diversity, I also don’t think you can conclude anything about the health of the soil based upon purely microbial richness.

Response: Those statements in regard to the effect of soil management practices (such as tillage and crop residue retention) on soil health have been modified to reflect what we really meant.

  1. Response to comment: Line 367: NTS had intermediate bacterial diversity; NT had the highest bacterial diversity. This sentence needs revising as based on incorrect conclusion.

Response: L426-427- We have made correction according to the reviewer’s comments.

  1. Response to comment: The PICRUSt authors suggest < 0.06 as a good match (and > 0.15 as a high one), I would suggest rephrasing that your matches were reasonable.

Response: This is a valuable point. We have rephrased the relevant statements. It is realized that the NSTI values for soil samples are often higher than those in mammalian guts based on the following references.

Langille, M.G.; Zaneveld, J.; Caporaso, J.G.; McDonald, D.; Knights, D.; Reyes, J.A.; Clemente, J.C.; Burkepile, D.E.; Thurber, R.L.V.; Knight, R.J.N.b. Predictive functional profiling of microbial communities using 16S rRNA marker gene sequences. 2013, 31, 814.

Wang, K.; Ye, X.; Zhang, H.; Chen, H.; Zhang, D.; Liu, L.J.S.r. Regional variations in the diversity and predicted metabolic potential of benthic prokaryotes in coastal northern Zhejiang, East China Sea. 2016, 6, 38709.

Round 2

Reviewer 2 Report

The authors have addressed most of my feedback appropriately and I appreciate the effort they have put in. I still have a couple of minor feedback points and I am concerned whether PICRUSt v1 can actually be used with taxonomy from Silva based upon reading the quality control help pages for the package so ideally would like someone who knows more about PICRUSt to comment upon that. I realise that my original feedback on ANOVA p-values was unclear, for which I apologise, but I would like to have the p-value for every ANOVA included in the manuscript not just those in table 3. Also, there are still a few references to bacterial species which need replacing:

Line 25: "observed species index"

Line 410: "profitable functional bacterial species" - perhaps replace with taxa or groups?

Line 384: "OUT" should be "OTU" - I presume autocorrected!

Author Response

  1. Response to comment: Add the p-value for every ANOVA included in the manuscript not just those in table 3.

Response: L189-196, 215-222, 292-295 - Following your instruction and suggestions, we have added the p-values for all ANOVA comparisons throughout the manuscript.

  1. Response to comment: Line 25: "observed species index" should be replaced with “OTU richness” in the section of Abstract.

Response: L25- we have revised the sentence and corrected the mistakes which the reviewer pointed out.

  1. Response to comment: Lines 410: "profitable functional bacterial species" - perhaps replace with taxa or groups?

Response: L406-We have made correction with the word “groups” according to the reviewer’s comments.

  1. Response to comment: Line 384: replace “OUT” with “OTU”, I presume autocorrected.

Response: L378- We are very sorry for our incorrect writing and have made correction.